# Research on the Quantitative Inversion of Soil Iron Oxide Content Using Hyperspectral Remote Sensing and Machine Learning Algorithms in the Lufeng Annular Structural Area of Yunnan, China

**DOI:** 10.3390/s24217039

**Published:** 2024-10-31

**Authors:** Yingtao Qi, Shu Gan, Xiping Yuan, Lin Hu, Jiankai Hu, Hailong Zhao, Chengzhuo Lu

**Affiliations:** 1School of Land and Resources Engineering, Kunming University of Science and Technology, Kunming 650093, China; 15758532473@163.com (Y.Q.); yxp_kust@163.com (X.Y.); 18508770991@163.com (J.H.); 15924918409@163.com (C.L.); 2Application Engineering Research Center of Spatial Information Surveying and Mapping Technology in Plateau and Mountainous Areas Set by Universities in Yunnan Province, Kunming 650093, China; 3School of Geography and Planning, Sun Yat-sen University, Guangzhou 510006, China; zhaohlong6@mail2.sysu.edu.cn

**Keywords:** soil, hyperspectral, iron oxide, characteristic wavelength selection, XGBoost model

## Abstract

This study used hyperspectral remote sensing to rapidly, economically, and non-destructively determine the soil iron oxide content of the Dinosaur Valley annular tectonic region of Lufeng, Yunnan Province. The laboratory determined the iron oxide content and original spectral reflectance (OR) in 138 surface soil samples. We first subjected the OR data to Savizky–Golay smoothing, followed by four spectral transformations—continuum removal reflectance, reciprocal logarithm reflectance, standard normal variate reflectance, and first-order differential reflectance—which improved the signal-to-noise ratio of the spectral curves and highlighted the spectral features. Then, we combined the correlation coefficient method (CC), competitive adaptive reweighting algorithm, and Boruta algorithm to screen out the characteristic wavelength. From this, we constructed the linear partial least squares regression model, nonlinear random forest, and XGBoost machine learning algorithms. The results show that the CC-Boruta method can effectively remove any noise and irrelevant information to improve the model’s accuracy and stability. The XGBoost nonlinear machine learning algorithm model better captures the complex nonlinear relationship between the spectra and iron oxide content, thus improving its accuracy. This provides a relevant reference for the rapid and accurate inversion of iron oxide content in soil using hyperspectral data.

## 1. Introduction

Iron is one of the most abundant metallic elements in the Earth’s crust and is of irreplaceable importance to production and life in human society. In nature, iron commonly occurs in a form resulting from chemosynthesis, such as iron ore and hematite; iron oxides are susceptible to environmental factors due to their high activity, and their occurrence cycle can reflect the process of soil leaching, the degree of weathering and development, oxidative precipitation, and migration, making it one of the important indexes for assessing the degree of soil development and for conducting soil classifications [1,2,3]. In addition, the iron oxide content in soil is a multifaceted and essential indicator that not only is useful in the assessment of soil fertility but also has critical scientific guidance and practical significance for monitoring environmental quality, guiding agricultural management, comprehending biogeochemical cycles, coping with climate change, preventing soil erosion, and restoring ecological soil [4,5]. Therefore, a quick grasp of information about the iron oxide content in soil can provide a scientific foundation for the sustainable development of agriculture, ecological and environmental protection, and rational utilization of land resources, thus promoting harmonious coexistence between human society and the natural environment.

The advancement of science and technology has led to an improvement in geochemical detection technology; this includes the utilization of inductively coupled plasma mass spectrometry, atomic emission spectrometry, atomic absorption spectrometry, and other conventional laboratory chemical detection methods to analyze the metal content in soil. These methods offer a low limit of detection value, high precision, and other advantages from the current mainstream operational detection methods [6]. However, the conventional detection method is constrained by its complex operation, high cost, and poor timeliness, which makes it challenging to realize the real-time and rapid acquisition of a wide range of soil information. The development of remote sensing technology has provided new methods in the fields of soil research and environmental monitoring that can quickly and efficiently obtain soil information on a large scale. In particular, hyperspectral remote sensing technology offers several advantages, including a high spectral resolution, spectral solid continuity, and the ability to perform multi-band simultaneous imaging. As a result, the soil spectral data collected through this technology provide abundant soil information that comprehensively reflects the current soil status [7,8]. In recent years, hyperspectral remote sensing technology has garnered significant attention from researchers who are conducting a range of studies in this area. This technique has the capability to capture real-time and dynamic information on soil physicochemical parameters. For example, Viscarra et al. [9] proposed that visible and near-infrared reflectance (VNIR) can be used to determine the content of iron oxides, clay minerals, carbonates, and organic matter in soil. Cheng et al. [10] demonstrated the feasibility of rapidly monitoring heavy metal concentrations in suburban soils using VNIR combined with a partial least squares regression (PLSR) model. Zhou et al. [11] used PLS, support vector machine (SVM), and random forest (RF) models to invert the concentrations of six heavy metals and iron/manganese oxides in the Three-River Source Region located in Southern Qinghai Province, China. The results showed that the RF model has good stability and predictions; its heavy metal determination coefficients are all greater than 0.83, and the determination coefficients of iron/manganese oxide in non-polluted areas are all greater than 0.88. Galvao et al. [12] showed that iron oxide absorption characteristics in soil lead to decreased reflectance in the visible band. He et al. [13] demonstrated a negative correlation between the iron oxide content and spectral reflectance in soil and that an increase in iron oxide content would lead to a decrease in spectral reflectance. Zhao et al. [14] performed conventional spectral transform and continuous wavelet transform (CWT) on soil spectral data, and the results showed that CWT could tap into the hidden spectral features and improve the correlation between soil spectral reflectance and iron oxide content. Ma et al. [15] performed a reflectance index transformation on an HJ-1A hyperspectral remote sensing image to develop a multiple linear regression (MLR) model that accurately inverts the free iron oxide content of surface soil. Xiong et al. [16] utilized the total reflection band to construct PLSR models for determining the concentrations of total, free, and amorphous iron in the soil, and the results show that free and amorphous iron models have small errors and that the iron silicate in total iron is susceptible to interference from organic matter and soil depth, which has a greater impact on the accuracy of the model. In addition, Table 1 visualizes the recent research progress of current researchers using soil spectral properties to predict iron, free iron, and iron oxide in soil.

In summary, many scholars have conducted numerous studies on the spectral characteristics of total iron, iron oxides, and some heavy metals, as well as their inversion models, and achieved favorable results, but most of them focused on the extensive research of linear regression models such as PLSR and MLR. With the rapid development of machine learning technology, which has been applied to a number of disciplines and has had a significant impact, especially when dealing with data with high dimensionality and complex nonlinear characteristics, machine learning shows excellent capabilities. Due to the numerous and complex nonlinear relationships between soil spectral data and metal content, as well as the differences in soil physicochemical properties in different regions, the results of previous studies need to be further verified. Therefore, the applicability of the results of previous studies to some specific regions needs to be further verified.

In this paper, the surface soil of the Lufeng Dinosaur Valley annulus in Yunnan Province is taken as the research object; firstly, the soil spectra are preprocessed to enhance the expression of information about iron oxide in soil; secondly, on the basis of the correlation coefficient method (CC), two variable preference algorithms, namely, competitive adaptive reweighted sampling (CARS) and the Boruta algorithm, are utilized for the selection of characteristic wavelengths, and the variables with random noise and low response relationships are removed. Finally, the linear PLSR model and the nonlinear RF and XGBoost two machine learning algorithm models are constructed. Using hyperspectral remote sensing technology combined with linear and nonlinear regression models for the quantitative inversion study of soil iron oxide content for comparison provides a scientific basis for the spectral analysis of hyperspectral soil iron oxide, the identification of characteristic wavelengths, and the selection of inversion models, which will help to deepen our understanding of the application of hyperspectral remote sensing technology in the monitoring of soil metal content and to develop such applications to be more efficient and accurate.

## 2. Materials and Methods

### 2.1. Study Area

The study area is located in Lufeng City, Chuxiong Yi Autonomous Prefecture, Yunnan Province, China (24°55′25″~24°58′00″ N,102°02′00″~102°05′30″ E). The area has a subtropical low-latitude plateau monsoon climate with typical characteristics: “no severe cold in winter, no scorching heat in summer, warm and rainy, dry and wet”. It also has obvious three-dimensional climate characteristics [14], an average historical annual temperature of 9–25 °C, and an average annual rainfall of 968.7 mm. The study area has a particular topographic structure, and the terrain is high all around and low in the middle. It is a typical small inland basin with its highest elevation at 2200 km and its lowest elevation at 1302 km. Lufeng City is an area with subtropical evergreen broad-leaved forest vegetation. According to the soil census data of Lufeng City in 1982 and 1985, there are 5 soil classes; 10 subclasses; and 20 soil genera, including brown loam, yellow-brown loam, red loam, purple loam, and paddy loam in the city; and 40 soil species [22].

### 2.2. Sample Collection and Data Acquisition

At the end of July 2021, soil samples were collected in the research region according to their topographic features, geomorphological conditions, and accessibility of the area. Each sample was collected following the 5-point mixed sampling method (range: 5 m × 5 m). The surface soil at depths of 5~20 cm was collected, and about 500 g was put into a sample bag, numbered, sealed, and preserved; a total of 138 soil samples were collected. The location of the study area and the distribution of sampling points are shown in Figure 1. In addition, the geographic coordinates of each sampling point and the surrounding environmental characteristics were recorded using devices such as a handheld GPS and cell phones.

After collecting the field data, we brought the soil samples back to the laboratory to naturally air-dry them, process them to remove impurities and stones, and then grind them through a 100-mesh sieve using an agate ball mill to minimize the effects of soil moisture, particle size, and impurities on the soil spectral data. When making measurements of soil spectral properties and iron oxide content, since these two measurements usually require the use of different instruments, the operation of which may cross-contaminate or cause interactions with the samples, separating the samples for measurement is an effective way to avoid interference. In addition, in order to ensure the accuracy of the experimental results, the ground soil samples need to be thoroughly mixed to ensure homogeneity so that even if the samples are divided into two batches for different purposes, the impact on the final experimental results will be small. Based on these considerations, we chose to split each soil sample after grinding into two parts: one for the measurement of the hyperspectral data of the soil and the other for the measurement of the iron oxide content in the soil. This experimental design helps to improve the accuracy of the measurements and the reliability of the experimental results. They were placed into round black vessels with a diameter of about 10 cm and a height of about 3 cm. The instrument used to measure the spectral data of the soil samples was an ASD Field Spec 3 (Analytical Spectral Devices, Inc. (ASD), Boulder, CO, USA). The spectrometer has a band range of 350 to 2500 nm; the light source was a built-in halogen light source probe with an inner diameter of 21 nm and a front field of view angle of 25°. The spectral sampling intervals were 1.4 nm (350~1000 nm) and 2 nm (1000~2500 nm). After resampling the spectral sampling interval to 1 nm, 2151 wavelengths were obtained [14]. Before measurement, the soil surface was scraped to reduce the effect of roughness on the spectral measurement results. When conducting spectral measurements in a dark room, it is important to avoid any objects that could influence the soil’s spectral reflectivity. This is done to minimize the impact of external environmental factors on the spectral measurement. Additionally, the instrument should be warmed up for 30 min, and the whiteboard should be optimized. The probe should be positioned approximately 2 cm above the soil surface. Five spectra were repeatedly collected in different directions for each sample, and the average value was taken as the actual spectral curve of the sample [22]. For the determination of iron oxide content, a NITON XL3t950 handheld ore element analyzer (NITON, USA) was used to determine the iron oxide content of the soil in strict accordance with the technical specifications for the chemical analysis of laboratory samples.

### 2.3. Data Preprocessing

Soil spectral curves are susceptible to random noise, which may come from instrumentation, environment, human operation, and other factors during data acquisition and measurement [23,24]. Random noise in spectral curves can significantly impact the precision and accuracy of the model, necessitating preprocessing of the collected hyperspectral data before modeling. The noisy bands at 300~399 nm and 2451~2500 nm were eliminated, and the main bands at 400~2450 nm with higher signal-to-noise ratios were retained to utilize the hyperspectral results for the inversion study of iron oxide content in soil. Furthermore, in order to prevent the negative impact on the modeling effect due to the existence of outliers in the iron oxide content data, an outlier elimination operation was performed, the box-and-line plot method was used to effectively identify and handle outliers, and three outliers were excluded by setting the judgement criteria to be plus or minus 1.5 times the quartile spacing, as shown in Figure 2.

To improve the signal-to-noise ratio further, highlight the spectral characteristics of the spectral data, and be able to retain the real spectral data while removing the effects of random noise, the original spectral reflectance (OR) was subjected to Savizky–Golay (SG) smoothing, with the number of windows set to 9 and the polynomial order set to 2. Because of the strong autocorrelation of the spectral data, different spectral transformations can be used to emphasize the characteristic differences in the spectra and thus enhance the response relationship between iron oxide content and spectral reflectance using four spectral transformations, namely, continuum removal reflectance (CR), reciprocal logarithm reflectance (RL), standard normal variate reflectance (SNV), and first-order differential reflectance (FD). These methods make spectral features more prominent by eliminating background noise, amplifying small variations, standardizing data distributions, and revealing curve slopes, which in turn improves the accuracy and reliability of soil analysis.

### 2.4. Characteristic Wavelength Selection

Due to the high dimensionality and redundancy of the hyperspectral data, using the entire band for modeling not only increases the amount of computation but also reduces model accuracy by considering bands with lower response relations. Therefore, it is crucial to select the optimal inversion band [22].

The correlation coefficient method (CC) is a correlation analysis of iron oxide content with the OR and various transformed spectral reflectances. The band that passes the *p* = 0.01 significance test is used as the characteristic wavelength; the higher the correlation, the stronger the sensitivity of the response [24,25,26]. The calculation formula is as follows:(1)$$\rho_{x,y}=\frac{cov(X,Y)}{\sigma_{x}\sigma_{y}}=\frac{m\sum_{i=1}^{m}x_{i}y_{i}-\sum_{i=1}^{m}x_{i}\sum_{i=1}^{m}y_{i}}{\sqrt{m\sum_{i=1}^{m}x_{i}^{2}-{\left(\sum_{i=1}^{m}x_{i}\right)}^{2}}\sqrt{m\sum_{i=1}^{m}y_{i}^{2}-{\left(\sum_{i=1}^{m}y_{i}\right)}^{2}}}$$
where X is the spectral reflectance, Y is the sample content (g.kg^−1^), $cov\left(X,Y\right)$ is the covariance of X and Y, $\sigma_{x},\;\sigma_{y}$ is the standard deviation of X and Y, m is the number of samples, $x_{i}$ is the spectral reflectance of sample i, and $y_{i}$ is the value of the content of sample i (g.kg^−1^).

Competitive adaptive reweighted sampling (CARS) is a variable selection method based on Darwin’s core idea of “survival of the fittest”. By iteratively combining the exponential decay function and the adaptive reweighted sampling technique, the bands with the largest absolute values of regression coefficients in the PLSR model are identified, and those with lower weights are excluded. The selected bands are then constructed into the PLSR model, the performance of the model is evaluated by 10-fold cross-validation, and the subset of bands with the smallest root-mean-square errors is filtered out as the optimal inversion bands [27,28]. The use of CC screening produces more characteristic bands, and further use of CARS can effectively reduce the variables with lower response relationships and obtain variables that are more beneficial to the modeling results. Due to the instability of the algorithm, in this study, the algorithm was run with 50 repetitions, and the frequency of occurrence of each wavelength was counted. Eventually, wavelengths with more than 20 frequencies were selected as characteristic wavelengths to ensure the reliability of the model, and the method was implemented through Matlab (R2023a) software.

The Boruta algorithm is an RF-based full correlation analysis screening feature algorithm for identifying and selecting the set of features that are relevant to the dependent variable. It is able to select the essential features from the original feature set while excluding the features that do not have a significant impact on the model predictions [29]. On the basis of obtaining the characteristic bands using CC, the Bourta algorithm can effectively remove some noise and irrelevant information. In this paper, the algorithm is implemented in Python (3.8.) using the ‘Brotutapy’ package, and the algorithm parameters are set as follows: ‘n_estimators’ is set to ‘auto’ to automatically select the number of estimators; ‘perc’ was set to 95 to determine the threshold of feature importance; ‘alpha’ was set to 0.05 for hypothesis testing; and’max_iter’ was set to 500 to specify the maximum number of iterations.

### 2.5. Modeling and Accuracy Evaluation

#### 2.5.1. Modeling

The PLSR is a new multiple regression analysis method proposed by Sven Wold and Carlos Albaro based on partial least squares, which combines the advantages of multiple linear regression analysis, correlation analysis, and principal component analysis [30,31]. PLSR can effectively deal with multiple covariate problems and handle high-dimensional data. In addition, it can project the high-dimensional spatial data onto the low-dimensional spatial data, which ensures accuracy and reliability of the model while reducing the complexity of the model. However, it also has drawbacks, such as a risk of overfitting, the limited ability to handle nonlinear relationships, and the sensitivity of parameter settings.

The RF algorithm is an integrated learning method that solves classification and regression problems by constructing multiple decision trees. In classification problems, each decision tree usually uses the Gini coefficient impurity or information gain as a criterion for data partitioning, while in regression problems, each node of each tree uses the mean or median of all the samples in the node as a predictive value [11,32,33]. In addition, RF randomly selects data and features when constructing each decision tree to improve the accuracy and generalization ability of the model, and the algorithm has the advantages of strong noise immunity, high robustness, and strong interpretability, which makes it suitable for dealing with high-dimensional, nonlinear relationship data.

The XGBoost algorithm is also an integrated learning algorithm that solves classification and regression problems by combining multiple decision trees [34]. Compared with the traditional gradient boosting tree, XGBoost introduces a regularization term, which not only helps to enhance the generalization ability of the model but also effectively controls its complexity. In terms of algorithm implementation, an efficient split-point finding algorithm is used, which can quickly and accurately locate the optimal split point. Moreover, XGBoost supports parallel computing, and through the update strategy of column blocks, efficient training can be realized on multiple machines, which significantly improves the ability to deal with large-scale datasets.

#### 2.5.2. Evaluation of Model Accuracy

The inversion model’s accuracy evaluation indexes include the coefficient of determination (R^2^); corrected root mean square error (RMSE_C_); validation root mean square error (RMSE_v_); and ratio of standard deviation (SD) to RMSE_v_, i.e., relative predictive deviation (RPD). R^2^ takes the value range of 0~1; the closer it is to 1, the better the effect of the model and the stronger the interpretability. RMSE_C_ is used to discriminate the prediction accuracy of the correction set on the samples; a smaller value indicates that the model is more accurate. RMSE_V_ is a reflection of the accuracy of the samples in the validation set; the closer the RMSE_C_ and the RMSE_V_ are, the more powerful the model’s stability. RPD is used to assess the model prediction performance of the important indicators [27]. In this paper, the RPD will be divided into three categories based on the predictive ability of the model: class A’s ability is better (RPD > 2.0), class B has a medium ability (1.4 < RPD < 2.0), and class C has a poor ability (RPD < 1.4).
(2)$$R^{2}=1-\frac{\sum_{i=1}^{m}{\left({\hat{y}}_{i}-\bar{y}\right)}^{2}}{\sum_{i=1}^{m}{\left(y_{i}-\bar{y}\right)}^{2}}$$
(3)$$RMSE_{c}=\sqrt{\frac{\sum_{i=1}^{m}{\left({\hat{y}}_{i}-y_{i}\right)}^{2}}{m-1}}$$
(4)$$RMSE_{v}=\sqrt{\frac{\sum_{i=1}^{m_{p}}{\left({\tilde{y}}_{i}-{\overset{̿}{y}}_{i}\right)}^{2}}{m_{p}-1}}$$
(5)$$RPD=\frac{SD}{RMSEP}$$
where ${\hat{y}}_{i}$ is the predicted value corresponding to sample point ⅈ; $y_{i}$ is the real value corresponding to sample point ⅈ; $\bar{y}$ is the average value of all sample points; and m is the number of samples. ${\tilde{y}}_{i}$ is the actual measured value of sample point i in the validation set, ${\overset{̿}{y}}_{i}$ is the predicted value obtained by establishing the regression model, and $m_{p}$ is the number of samples in the validation set.

The workflow of this study is shown in Figure 3.

## 3. Results and Analysis

### 3.1. Statistical Characterization of Soil Iron Oxide Content

After removing outliers, we statistically analyzed the remaining 135 soil samples, dividing them into a 7:3 ratio using the Kennard–Stone algorithm, with 94 samples in the calibration set and 41 in the validation set. The iron oxide content in the study area ranged from 18.293 to 66.978 g.kg^−1^ with a mean value of 41.201 g.kg^−1^ and a coefficient of variation (CV) of 28.4%. Soil science classifies soil properties based on the CV, ranging from 0 to 15% for low variation, 16% to 36% for moderate variation, and exceeding 36% for high variation [35]. The CV of the iron oxide content of the soil in the study area is between 16% and 36%, which is moderate variability, indicating that the distribution of iron oxide content in the study area has a certain spatial variability. Table 2 displays the statistical results.

### 3.2. Characterization of Soil Spectral Profiles

Although the overall spectral reflectance of all the samples is low, ranging from 0 to 0.5, the trends are generally similar. In the visible spectral band (400~700 nm), the growth trend is similar to a steep hill; the reflectance with the increase in wavelength increases rapidly; its spectral reflectance growth rate is the most significant in all the wavelength ranges; and the reflectance increases at a slower rate between 700 and 1000 nm, remains relatively stable between 1100 and 1900 nm, and displays a decreasing tendency between 1900 and 2450 nm. Different soil components produced different absorption features in the corresponding spectral bands. The absorption effect of iron/manganese oxides in the soil mainly affected weak absorption peaks near the spectral bands of 500 nm and 900 nm. Near 1400 nm, the water molecule hydroxyl group OH in silicate minerals underwent contraction and vibration, forming a clear absorption valley. At about 1900 nm, there was a strong absorption valley caused by water vapor absorption. At 2200 nm, there was an absorption valley caused by the contraction vibration effect of the water molecule OH in clay minerals [11,34]. The iron oxide content of the soil was classified from high to low and divided into six intervals. The average iron oxide content and its corresponding spectral reflectance were calculated for each interval, and the spectral curves corresponding to different iron oxide contents were plotted to understand better the spectral characteristics corresponding to different iron oxide contents. The findings demonstrated that the spectral reflectance and iron oxide concentration were negatively correlated, with the greater the iron oxide content, the lower the spectral reflectance (in Figure 4).

SG smoothing of the OR eliminates some of the random noise and interfering information, but there is still interfering information due to stray light and the background. Therefore, the spectral transformation of the OR is needed to highlight the spectral characteristics. The CR, RL, SNV, and FD transforms were performed on the OR, and the transformation results are shown in Figure 5. After applying CR processing, the original spectral curves were adjusted to have a consistent background, which effectively emphasized the absorption features of the spectra. This resulted in clear absorption peaks at 500 nm, 900 nm, 1400 nm, 1900 nm, 2200 nm, and 2400 nm, respectively. The RL transformation of the original spectral data increased the reflectance of the visible wavelength. The SNV spectral transformation preserved some of the spectral absorption features and increased the slope of the spectral curve in the 400–900 nm range. The FD transformation enhanced the sensitivity of the original spectral curve, revealing more spectral details that could not be captured by the other three spectral transformations.

### 3.3. Correlation Analysis between Spectra and Iron Oxide Content

According to the Pearson correlation analysis, different spectral reflectances have varying effects on the correlation of iron oxide content, as shown in Figure 6. OR is negatively correlated with iron oxide content, and its correlation coefficient reaches its maximum value at 403 nm (r = −0.555). RL is positively correlated with iron oxide content, and its correlation coefficient reaches its maximum value at 436 nm (r = 0.596). The correlation coefficients of SNV, FD, and CR fluctuate more than the previous two spectral transforms and reach their maximum values, respectively, at the 2127 nm (r = −0.631), 1365 nm (r = −0.533), and 1883 nm (r = −0.634) wavelengths. The absolute value of the maximum correlation coefficient of RL increased by 0.046 compared with OR, which indicates that RL lessens the impact of random noise and also eliminates part of the influence of cloudy scattering. SNV increased by 0.076 compared with OR, which suggests that SNV is effective in decreasing the influence of the size of the solid particles, the surface scattering, and the change in the optical range on the spectra. CR increased by 0.079 compared with OR, and CR effectively highlights the absorption features and reflection features. The three spectral transformations effectively enhance the correlation with iron oxide content. However, the maximum correlation coefficient of FD is reduced by 0.024 compared with OR, which indicates that the differential transform reduces the interference of the background, while the increased high-frequency noise is an important reason for the reduced correlation. The correlation between the spectral data after FD, SNV, and CR spectral transform processing and the iron oxide content is not limited to a single positive or negative correlation, and this alternating positive and negative correlation reveals that FD, SNV, and CR spectral transform processing can effectively enhance the spectral data and provide a reliable basis for the establishment of an accurate quantitative analysis model.

### 3.4. Selection of Characteristic Bands

The characteristic band selection based on the CC was carried out with the help of a Pearson correlation analysis of the iron oxide content using OR, and the four transformed spectral reflectances and the number of bands in which OR, CR, FD, RL, and SNV passed the *p* = 0.01 significance test were 2051, 1425, 1318, 2051, and 1737, respectively. The absolute value of the correlation coefficient’s critical value to reach a significant correlation (*p* = 0.01) is 0.2613. Soil iron oxide content was significantly correlated with both OR and RL, so the whole band passed the significance test; the significance bands of iron oxide content and CR were mainly concentrated near 400~600 nm, 1200~1900 nm, 2100 nm, and 2300 nm; FD was mainly concentrated in the areas of 400~500 nm, 1000~1700 nm, and 2000 nm; and SNV was mainly concentrated near 400~550 nm, 600~1300 nm, and 1400~2300 nm.

The direct use of variable preference algorithms over the full bandwidth may reduce the modeling accuracy due to inefficiencies and the omission of critical information. The CC is instructive for assessing the linear correlation between iron oxide content and spectral reflectance, but its application is limited when analyzing in depth the significance of different band characteristics. Therefore, in this paper, we first use the correlation analysis method to preliminarily screen out any sensitive bands and then use CARS and the Bourta algorithm to further screen out any sensitive characteristic bands that have a greater contribution to the inversion model.

As can be seen in Figure 7, the number of bands further selected using the optimization algorithm is significantly reduced compared to the characteristic bands obtained with the CC. The bands screened by CARS are mainly concentrated in the visible and short-wave infrared bands. The Bourta algorithm effectively eliminates some of the noise and irrelevant information and, therefore, screens a smaller number of feature bands compared to CARS. The Bourta algorithm selectively filters out the distinctive bands located at 500 nm and 900 nm due to the influence of iron/manganese oxides’ absorption. Additionally, it also excludes the bands near 1900 nm due to interference caused by water absorption.

### 3.5. Inversion Model Construction and Accuracy Evaluation

In this study, the models constructed using PLSR, RF, and XGBoost for estimating iron oxide content in soil were implemented in the Python third-party library scikit-learn, and parameter optimization was performed using the learning curve method. In order to determine the optimal parameter configuration for each model, a ten-fold cross-validation method was used to evaluate the model performance under different parameter settings, and the parameter combination with the smallest RMSE_V_ and the highest coefficient of determination, $R_{V}^{2}$, was selected. The important parameter settings for each model are shown in Table 3.

#### 3.5.1. Analysis of PLSR Model Prediction Results

The PLSR model inversion results are shown in Table 4. Among the PLSR models, the accuracy of the validation set of the RL-CC-CARS-PLSR model is the highest among the models constructed using the CC-CARS algorithm to screen the characteristic wavelengths, in which the $R_{V}^{2}$, RMSE_V_, and RPD of the validation set are 0.720, 7.019, and 1.891, respectively, and the accuracy of the validation set of the CR-CC-Boruta-PLSR model is the highest among the models constructed using the CC-Boruta algorithm to screen the characteristic wavelengths. The Boruta-PLSR model has the highest accuracy, where the $R_{V}^{2}$, RMSE_V_, and RPD of the validation set are 0.613, 8.258, and 1.607, respectively. The two best PLSR models have RPDs between 1.4 and 2, and the models have medium predictive power.

#### 3.5.2. Analysis of RF Model Prediction Results

Table 5 displays the inversion results of the RF model. In the RF model, comparing the modeling effects of the two algorithms, CC-Boruta and CC-CARS, for screening the characteristic bands, the modeling accuracy of the characteristic bands screened by the CC-Boruta algorithm is higher than that of the CC-CARS algorithm. The model CR-CC-CARS-RF has the highest accuracy on the validation set compared to other models developed with the involvement of CC-CARS. It has $R_{V}^{2}$ and RPD values of 0.481 and 1.508, respectively. Compared with the CR-CC-Boruta-RF model constructed with the participation of CC-Boruta, the $R_{V}^{2}$ and RPD of the model decreased by 0.209 and 0.337, respectively. The CR-CC-Boruta-RF model inversion outperformed all other models. It achieved a validation set $R_{V}^{2}$ value of 0.690, an RMSE_V_ value of 6.264 g.kg^−1^, and an RPD value of 1.845. The model’s predictive performance was considered average, as its RPD value fell between 1.4 and 2.0.

#### 3.5.3. Analysis of XGBoost Model Prediction Results

Table 6 shows that the CR-CC-CARS-XGBoost model inversion performs better in the XGBoost model based on the CC-CARS algorithm for selecting the characteristic bands, with the validation sets’ $R_{V}^{2}$, RMSE_V_, and RPD being 0.675, 6.594 g.kg^−1^, and 1.753, respectively. The CC-Boruta algorithm selects the characteristic bands to build the CR-CC-Boruta-XGBoost and FD-CC-Boruta-XGBoost model inversions with the best results, and its validation set $R_{V}^{2}$ is 0.777 and 0.806, RMSE_V_ is 5.464 g.kg^−1^ and 5.087 g.kg^−1^, and RPD is 2.116 and 2.272, respectively.

#### 3.5.4. Model Accuracy Comparison

Comparing the performance of the three inversion models, the PLSR models CR\RL-CC-CARS-PLSR and CR-CC-Boruta-PLSR performed relatively well in predicting the iron oxide content of soils, although they did not excel in overall predictive performance. In the comparison of RF models, the CR-CC-CARS-RF and CR\SNV\FD-CC-Boruta-RF models had average predictive performances, while the remaining models had poor predictive performances. Among the XGBoost models, the CR\FD-CC-Boruta-XGBoost model shows a better prediction effect, with RV2 exceeding 0.7 for all of them, while the CR\RL\SNV\CC-CARS-XGBoost and SNV\CC-Boruta-XGBoost models show average predictive abilities and the rest of the models are relatively weak. In order to demonstrate more intuitively the performance of the PLSR, RF, and XGBoost models in predicting the iron oxide content in the soil of the study area, the fitted plots of the predicted and real values of the four models—CR-CC-CARS-PLSR, CR-CC-Boruta-RF, and CR\FD-CC-Boruta-XGBoost—were plotted as shown in Figure 8. The predicted and true values of these models were closely distributed around the 1:1 line, indicating that they have the potential to serve as valid models for predicting iron oxide content in soil. By comprehensively evaluating the accuracies of the modeling and validation sets, the FD-CC-Boruta-XGBoost model was considered the best model for the inversion of iron oxide content in the soil in this study area due to its high prediction accuracy and stability.

## 4. Discussion

Focusing on the Dinosaur Valley annulus in Lufeng City, Chuxiong Yi Autonomous Prefecture, Yunnan Province, China, this study aims to quantitatively invert the iron oxide content in soil using soil spectroscopy. It was shown that OR was negatively correlated with iron oxide content, and the higher the iron oxide content in the soil, the lower the original spectral reflectance, which was consistent with the research results of He et al. [13]. The four common spectral transforms (CR, RL, SNV, and FD) used in this paper improved the response relationship between soil iron oxide content and spectral reflectance to some degree. However, fractional-order differential and wavelet transforms have been widely used in soil hyperspectral analysis and have shown better results [36,37,38,39,40,41,42]. Therefore, the next study will use fractional order differentiation and wavelet transform to mine more spectral information to achieve higher inversion accuracy.

Due to the high amount of redundant information in hyperspectral data, it is very necessary to carry out characteristic spectrum selection. In this paper, based on the CC, two variable preference algorithms, CARS, and Bourta, were used for the selection of feature bands. The results show that the Bourta algorithm can screen out the variables with stronger response relationships and provide a better-quality basis for modeling, which is in line with the results of the study by Luo, Zhang, and Mao et al. [29,43,44]. CC is not suitable for the selection of characteristic bands in soils with low concentrations and weak correlations for some metal elements. Therefore, the applicability of the method used in previous studies to some special cases requires further validation and research.

Considering the limitations of the special topographic structure, area scale, sampling point deployment, and sample volume of this study area, the applicability of the hyperspectral remote sensing quantitative inversion model of iron oxides in soils established in this paper to other areas needs to be further verified [24]. Meanwhile, this study used non-imaging spectral inversion; the application of a large-scale range may be limited; therefore, combining satellite or airborne acquisition of imaging spectra for large-scale regional inversion is the trend of future research [45,46].

## 5. Conclusions

(1)The spectral transformation of OR using CR, RL, SNV, and FD further determined the correlation between soil iron oxides and spectra: OR was negatively correlated with soil iron oxide content; RL was positively correlated with iron oxide content; and CR, RL, and SNV could improve the correlation coefficients. In addition, the FD transformation improved the sensitivity of the spectral curves and effectively highlighted more detailed spectral characteristics, thus positively affecting the prediction accuracy of the model.(2)The selection of characteristic bands contributes positively to improving the model’s accuracy. The CC-Boruta algorithm can effectively remove some random noise and interference information. The feature band selection is small and targeted, which, to some extent, minimizes the optimal combination of features and has a better effect on simplifying the model and improving its stability. In addition, it is worth noting that the CC-CARS method significantly outperforms the CC-Boruta method in capturing the characteristic wavelengths of the linear relationship between the spectra and the iron oxide content.(3)Soil spectra are affected by many factors, among which iron oxide content and spectra contain both linear and nonlinear relationships. In this paper, by comparing the PLSR linear regression model and two machine learning nonlinear models—RF and XGBoost—the machine learning algorithms can better express the nonlinear relationship and effectively improve the inversion accuracy. Therefore, nonlinear inversion algorithms such as machine learning can be prioritized when using the hyperspectral inversion of iron oxide content in soil.(4)In the hyperspectral iron oxide inversion model in soil constructed in this study, the $R_{V}^{2}$ of the CR-CC-Boruta-XGBoost model is 0.777 and its RMSE_V_ is 5.464, and the $R_{V}^{2}$ of the FD-CC-Boruta-XGBoost model is 0.806 and its RMSE_V_ is 5.087. The RPDs of the two models are 2.116 and 2.272, both exceeding 2.0, indicating that both models are able to effectively estimate the iron oxide content in soil in the study area. The $R_{V}^{2}$ of the FD-CC-Boruta-XGBoost model was more than 0.8, which was the best inversion model for the present study, and it was able to more accurately invert the iron oxide content in the soil as compared with the other models.

## Figures and Tables

**Figure 1 sensors-24-07039-f001:**
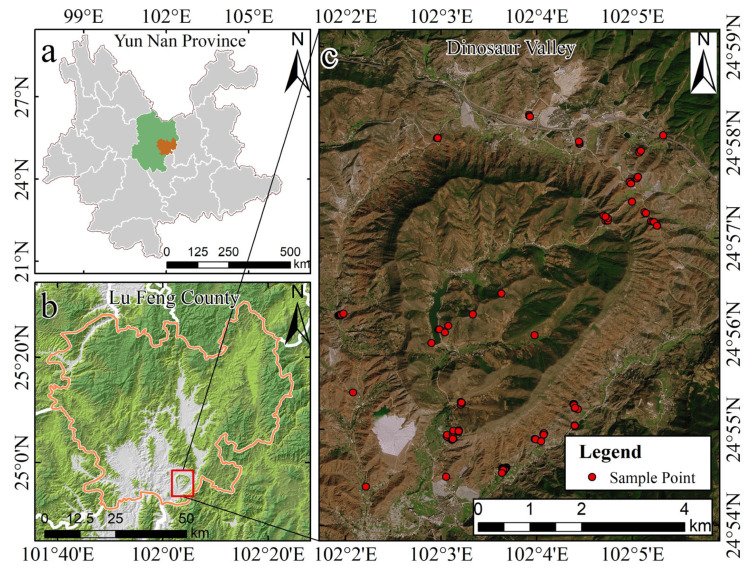
Location of the study area and distribution of sampling points. (**a**) Yunnan Province, China; (**b**) Lufeng City, Chuxiong Yi Autonomous Prefecture; (**c**) Dinosaur Valley.

**Figure 2 sensors-24-07039-f002:**
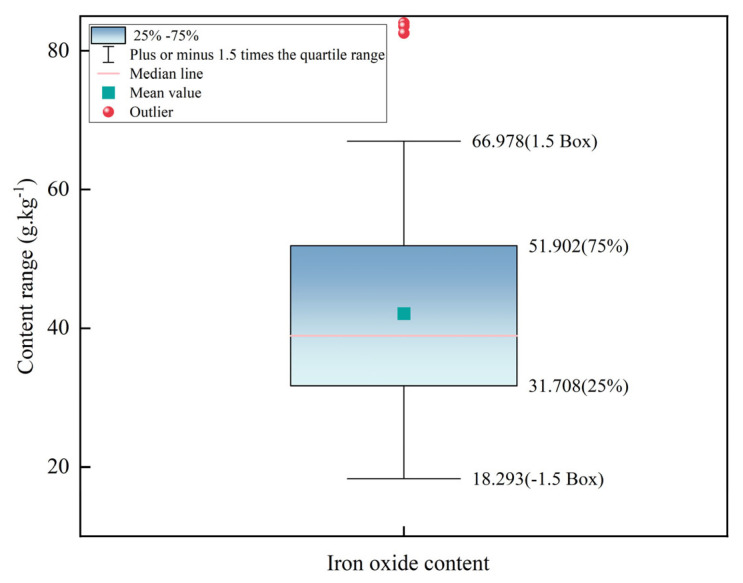
Box plot of iron oxide content.

**Figure 3 sensors-24-07039-f003:**
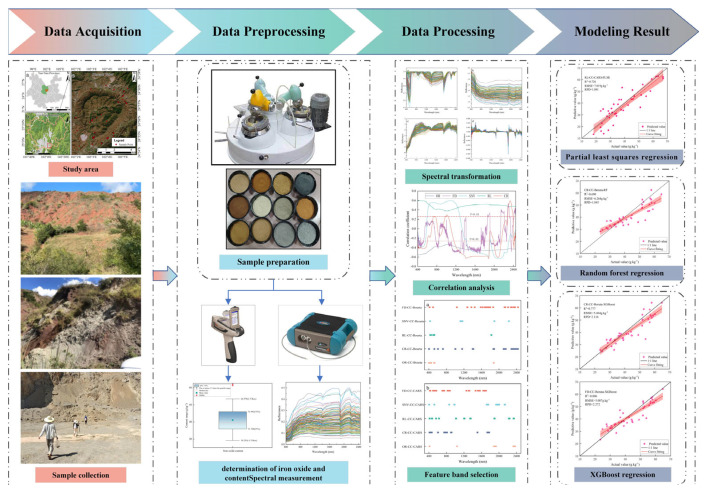
Research workflow diagram.

**Figure 4 sensors-24-07039-f004:**
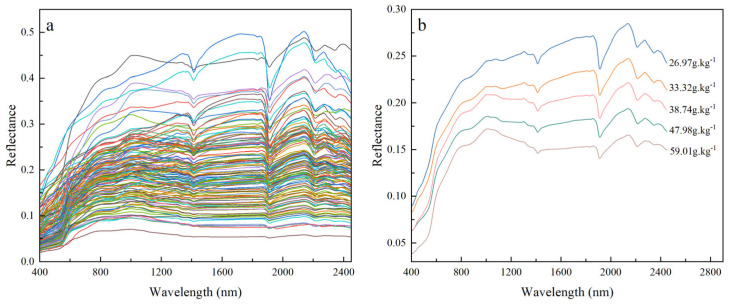
(**a**) Original spectral curves of soil samples; (**b**) average spectral reflectance after six equal divisions of the original spectral curve.

**Figure 5 sensors-24-07039-f005:**
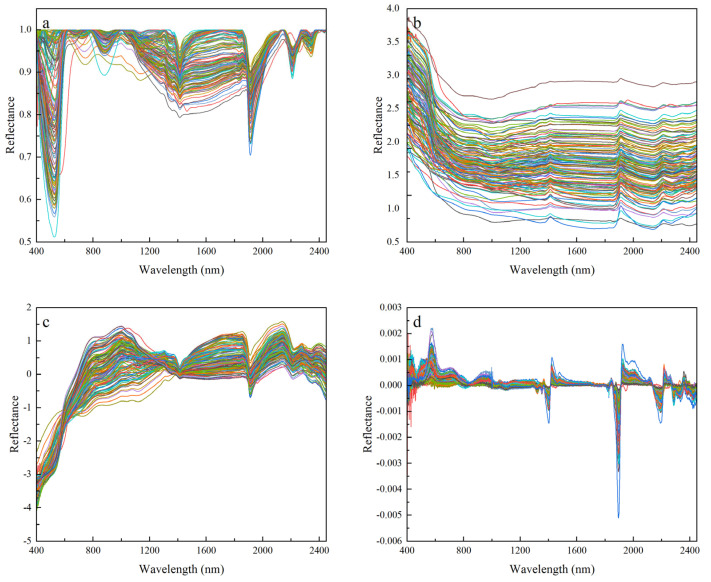
Spectral curve after transforming the original spectrum. (**a**) CR; (**b**) RL; (**c**) SNV; (**d**) FD.

**Figure 6 sensors-24-07039-f006:**
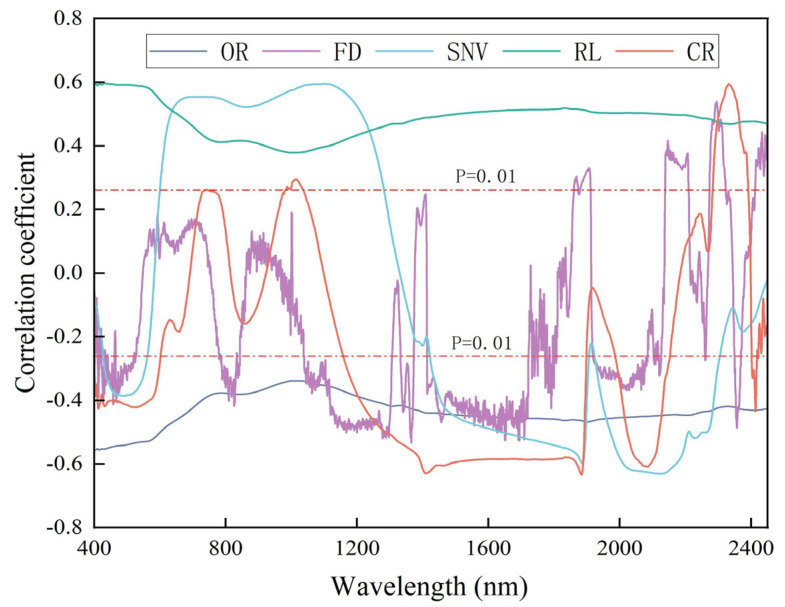
Correlation coefficients between the original spectrum and its transformed spectral reflectance and iron oxide content.

**Figure 7 sensors-24-07039-f007:**
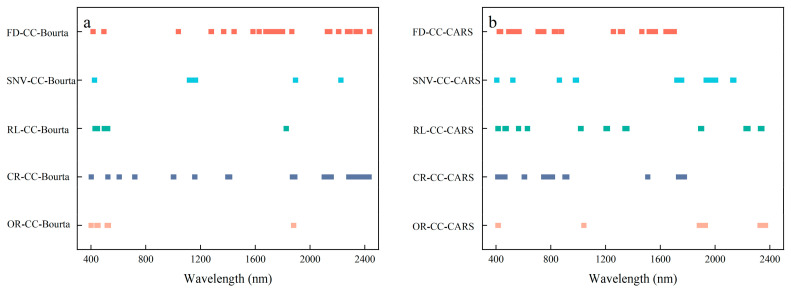
Characteristic band selection results. (**a**) CC-Bourta; (**b**) CC-CARS.

**Figure 8 sensors-24-07039-f008:**
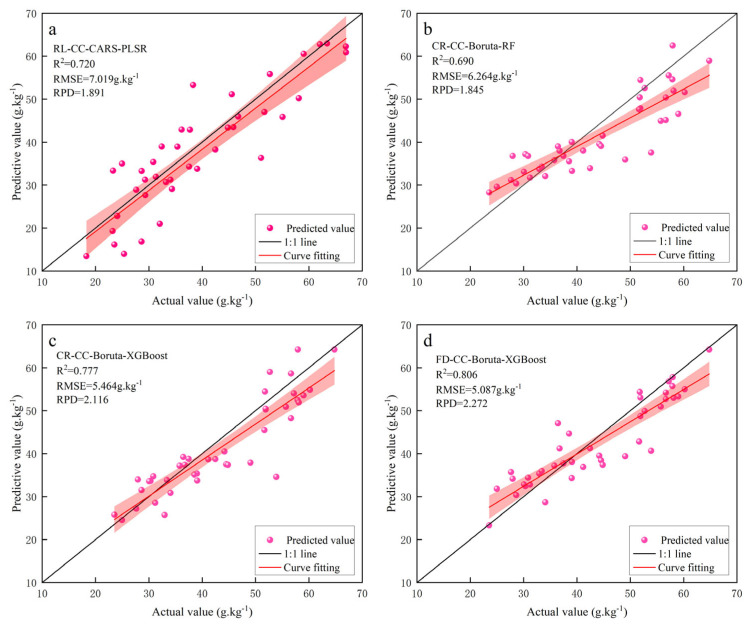
Fitting plot of true values to predicted values. (**a**) CR-CC-CARS-PLSR; (**b**) CR-CC-Boruta-RF; (**c**) CR-CC-Boruta-XGBoost; (**d**) FD-CC-Boruta-XGBoost.

**Table 1 sensors-24-07039-t001:** A summary of recent applications of the use of soil spectral properties for the prediction of iron, free iron, and iron oxide in soils. ^A^: well predicted (R^2^ > 0.8), ^B^: acceptable prediction (0.6 < R^2^ < 0.8), ^C^: poor prediction (R^2^ < 0.6).

Model	Sample Size	Predicted Properties	References
PLSR	93	Fe ^A^	[10]
MLR	174	Fe_2_O_3_ ^C^	[13]
SVMR	135	Fe_2_O_3_ ^B^	[14]
MLR	82	Fe_2_O_3_ ^A^	[15]
PLSR	160	Fe ^A^. free iron ^B^. Fe_2_O_3_ ^B^	[16]
PLSR	95	free iron ^B^	[17]
PLSR	36	Fe ^A^	[18]
PLSR	255	Fe ^C^	[19]
PLSR	146	Fe_2_O_3_ ^B^	[20]
SVMR	592	Fe_2_O_3_ ^A^	[21]

**Table 2 sensors-24-07039-t002:** Statistical characteristics of soil iron oxide content in the study area.

Dataset	Range (g.kg^−1^)	Mean (g.kg^−1^)	Standard Deviation (g.kg^−1^)	Variable Coefficient (%)
Total dataset (*n* = 135)	18.293~66.978	41.201	11.698	28.4%
Calibration dataset (*n* = 94)	18.293~66.978	40.205	11.617	28.9%
Validation dataset (*n* = 41)	23.534~64.808	43.483	10.593	23%

**Table 3 sensors-24-07039-t003:** Parameter settings of the model.

Model	Hyperparameters	Range of Values
PLSR	n_components	[1, 20]
RF	n_estimators	[30, 200]
	max_depth	[2, 16]
	max_features	[5, 29]
	min_samples_leaf	[1, 37]
	min_samples_split	[2, 21]
XGBoost	n_estimators	[50, 100]
	subsample	[0.1, 0.7]
	max_depth	[1, 10]
	learning_rate	[0.07, 0.49]
	gamma	[0, 8]

**Table 4 sensors-24-07039-t004:** Results of PLSR inversion for iron oxide content in soil.

CC	Wavelength Selection	Quantities	Calibration Set		Validation Set		
			$R_{C}^{2}$	RMSE_C_ (g.kg^−1^)	$R_{V}^{2}$	RMSE_V_ (g.kg^−1^)	RPD
OR	CC-CARS	27	0.687	6.002	0.375	10.490	1.265
CR		52	0.625	6.564	0.640	7.968	1.666
RL		62	0.816	4.603	0.720	7.019	1.891
SNV		68	0.680	6.066	0.404	10.243	1.296
FD		53	0.510	7.508	0.223	11.694	1.135
OR	CC-Boruta	11	0.486	7.689	0.176	12.049	1.101
CR		70	0.545	7.236	0.613	8.258	1.607
RL		13	0.550	7.192	0.347	10.722	1.238
SNV		30	0.329	8.786	0.414	10.155	1.307
FD		53	0.587	6.894	0.585	8.552	1.552

**Table 5 sensors-24-07039-t005:** Results of RF inversion for iron oxide content in soil.

CC	Wavelength Selection	Quantities	Calibration Set		Validation Set		
			$R_{C}^{2}$	RMSE_C_ (g.kg^−1^)	$R_{V}^{2}$	RMSE_V_ (g.kg^−1^)	RPD
OR	CC-CARS	27	0.891	9.183	0.239	9.819	1.177
CR		52	0.921	5.228	0.481	7.663	1.508
RL		62	0.882	11.556	0.260	12.016	0.926
SNV		68	0.923	3.099	0.252	9.752	1.185
FD		53	0.914	11.556	0.399	12.009	0.963
OR	CC-Boruta	11	0.900	8.646	0.351	9.397	1.230
CR		70	0.934	2.850	0.690	6.264	1.845
RL		13	0.905	11.409	0.363	11.914	0.970
SNV		30	0.925	4.608	0.591	6.856	1.686
FD		53	0.943	5.547	0.699	7.086	1.631

**Table 6 sensors-24-07039-t006:** XGBoost inversion results of iron oxide content in soil.

Spectral Type	Wavelength Selection	Quantities	Calibration Set		Validation Set		
			$R_{C}^{2}$	RMSE_C_ (g.kg^−1^)	$R_{V}^{2}$	RMSE_V_ (g.kg^−1^)	RPD
OR	CC-CARS	27	0.896	3.724	0.433	8.707	1.328
CR		52	0.827	4.804	0.675	6.594	1.753
RL		62	0.894	3.756	0.508	8.111	1.425
SNV		68	0.965	2.166	0.528	7.944	1.455
FD		53	0.916	3.347	0.457	8.519	1.357
OR	CC-Boruta	11	0.812	5.009	0.424	8.770	1.318
CR		70	0.972	1.944	0.777	5.464	2.116
RL		13	0.741	5.881	0.395	8.993	1.250
SNV		30	0.865	4.247	0.650	6.840	1.690
FD		53	0.914	3.382	0.806	5.087	2.272

## Data Availability

The data presented in this study are available from the corresponding author on request.

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
