# Peer review of "Research on the Quantitative Inversion of Soil Iron Oxide Content Using Hyperspectral Remote Sensing and Machine Learning Algorithms in the Lufeng Annular Structural Area of Yunnan, China"

_sensors, 2024, doi:10.3390/s24217039_

Round 1

Reviewer 1 Report

Comments and Suggestions for Authors

In this paper, hyperspectral inversion of ferric oxide content in soil is of some significance to the application of hyperspectral in ore quantification, but some experimental steps and meanings are not clearly expressed in this paper.Here are the detailed recommendations:

(1)The characteristic wavelength in this paper is selected based on CC. Please indicate the full name when using CC for the first time, so as not to confuse readers.

(2)The accuracy of a model is always comprehensively affected by multiple factors. To compare RF and XGBoost two models, although the training samples are well controlled and consistent, there are still many parameter Settings of the model in actual training that will affect the final accuracy, but the method and basis of parameter setting of the two models are not explained in this paper.

Author Response

For research article

Response to Reviewer Comments

  1. Summary

I would like to thank you for your time and effort in reviewing my thesis. Your valuable comments have been very helpful in improving my thesis. The following is a point-by-point response to your comments:

2. Questions for General Evaluation

Reviewer’s Evaluation

Response and Revisions

Does the introduction provide sufficient background and include all relevant references?

Can be improved

The second part of the introduction presents in tabular form the methods and precision used in some of the most recent research advances in the field to date.

Is the research design appropriate?

Yes

A linear partial least squares model was added to the paper to compare with a non-linear machine learning algorithm model to highlight the superiority of machine learning algorithms.

Are the methods adequately described?

Yes

The methods and parameter settings used in this paper were further modified.

Are the results clearly presented?

Can be improved

Added linear partial least squares model made corresponding improvements in the conclusion section.

Are the conclusions supported by the results?

Yes

Corresponding changes have been made.

  1. Point-by-point response to Comments and Suggestions for Authors

Comments 1: The characteristic wavelength in this paper is selected based on CC. Please indicate the full name when using CC for the first time, so as not to confuse readers.

Response 1: Thank you for pointing this out. We agree with this comment. Therefore, I I've already made changes in the paper. On page 6, line 215, the specific modifications are as follows:

The correlation coefficient method (CC) is a correlation analysis of iron oxide content with the OR and various transformed spectral reflectances. The band that passes the P=0.01 significance test is used as the characteristic wavelength; the higher the correlation, the stronger the sensitivity of the response [24-26].

Comments 2: The accuracy of a model is always comprehensively affected by multiple factors. To compare RF and XGBoost two models, although the training samples are well controlled and consistent, there are still many parameter Settings of the model in actual training that will affect the final accuracy, but the method and basis of parameter setting of the two models are not explained in this paper.

Response 2: Thank you for pointing this out. We agree with this comment. The corresponding chapter on parameterization has been adjusted and has now been moved to the chapter “Inversion model construction and accuracy evaluation,"  and the parameterization methods and rationale for the three models are explained. On page 11, line 383, the specific modifications are as follows:

In this study, the models constructed using PLSR, RF, and XGBoost for estimating iron oxide content in soil were implemented in the Python third-party library scikit-learn, and parameter optimization was performed using the learning curve method. In order to determine the optimal parameter configuration for each model, a ten-fold cross-validation method was used to evaluate the model performance under different parameter settings, and the parameter combination with the smallest RMSEV and the highest coefficient of determination,R_V^2, was selected, and the important parameter settings for each model are shown in Table 3.

Table 3. Parameter settings of the model

Model

Hyperparameters

Range of values

PLSR

n_components

[1,20]

RF

n_estimators

[30,200]

max_depth

[2,16]

max_features

[5,29]

min_samples_leaf

[1,37]

min_samples_split

[2,21]

XGBoost

n_estimators

[50,100]

subsample

[0.1,0.7]

max_depth

[1,10]

learning_rate

[0.07,0.49]

gamma

[0,8]

  1. Response to Comments on the Quality of English Language

Point 1:  NO

Response 1: Changes were made to address grammatical issues in the paper accordingly.

  1. Additional clarifications

The following areas have been added to this revision:

  • Changes were made to the title of the paper
  • The second part of the introduction presents in tabular form the methods and precision used in some of the most recent research advances in the field to date.
  • A linear partial least squares model was added to the paper to compare with a non-linear machine learning algorithm model to highlight the superiority of machine learning algorithms.
  • Key details such as parameterization of the two characteristic bands' variable preference methods are discussed.The parameter optimization methods and principles of the three models are discussed.

Reviewer 2 Report

Comments and Suggestions for Authors

Author Response

For research article

Response to Reviewer Comments

  1. Summary

I would like to thank you for your time and effort in reviewing my thesis. Your valuable comments have been very helpful in improving my thesis. The following is a point-by-point response to your comments:

2. Questions for General Evaluation

Reviewer’s Evaluation

Response and Revisions

Does the introduction provide sufficient background and include all relevant references?

Can be improved

The second part of the introduction presents in tabular form the methods and precision used in some of the most recent research advances in the field to date.

Is the research design appropriate?

Must be improved

Linear modeling added at the suggestion of the reviewers

Are the methods adequately described?

Must be improved

The methods and parameter settings used in this paper were further modified.

Are the results clearly presented?

Can be improved

Added linear partial least squares model made corresponding improvements in the conclusion section.

Are the conclusions supported by the results?

Can be improved

Corresponding changes have been made.

  1. Point-by-point response to Comments and Suggestions for Authors

Comments 1: It is recommended to present the research progress on the second page in the form of a table, summarizing the methods used and the corresponding accuracy. This approach will facilitate a clearer understanding for the readers.

Response 1: Thank you for pointing this out. We agree with this comment. The importance of presenting research advances in a tabular format, summarizing the methods used and the corresponding accuracy, really helps the reader to have a clearer understanding of our research methods and accuracy. Therefore, I have added in the introduction section of the paper the latest research progress of current researchers using soil spectral properties to predict iron, free iron, and iron oxide in soil. The specific results are given below:

Table 1. A summary of recent applications of the use of soil spectral properties for the prediction of iron, free iron, and iron oxide in soils. A: well predicted (R2 > 0.8), B: acceptable prediction (0.6 < R2 < 0.8), C: poor prediction (R2 < 0.6). (On page 4, line107)

Model

Sample size

Predicted properties

References

PLSR

93

FeA

[10]

MLR

174

Fe2O3C

[13]

SVMR

135

Fe2O3B

[14]

MLR

82

Fe2O3A

[15]

PLSR

160

FeA. free ironB. Fe2O3B

[16]

PLSR

95

free ironB

[17]

PLSR

36

FeA

[18]

PLSR

255

FeC

[19]

PLSR

146

Fe2O3B

[20]

SVMR

592

Fe2O3A

[21]

Comments 2: The paper title only mentions the XGBoost algorithm, but the manuscript discusses both Random Forest (RF) and XGBoost algorithms, leading to an inconsistency between the title and the research content. Therefore, it is recommended to modify the title to reflect the use of both algorithms, aligning it with the research content in the manuscript. Alternatively, you may choose to retain only the discussion related to the XGBoost algorithm in the manuscript to ensure consistency between the title and the manuscript content.

Response 2: Thank you for pointing this out. We agree with this comment. The main purpose of this paper is to highlight the superiority of machine learning algorithm models combined with hyperspectral remote sensing to invert the iron oxide content in soil, and the title of the paper is adjusted according to the reviewer's comments, and is now changed to “Research on the quantitative inversion of soil iron oxide content using hyperspectral remote sensing and machine learning algorithms in the Lufeng Annular Structural Area of Yunnan, China.”.

Comments 3: "On page 3, lines 138-143, it states, 'After grinding, each sample was divided into two parts, one to determine the soil hyperspectral data and the other to determine the

soil iron oxide content,' but no explanation is provided for why this method of measurement was used. Why not measure both the spectral reflectance and the iron content from the same sample? Dividing the samples in this manner could lead to inconsistencies between the spectral reflectance and iron content, introducing potential errors. This makes it difficult to understand the rationale behind this approach. Please provide a clear explanation for the necessity of splitting the samples for measurement."

Response 3: Thank you for pointing this out. We agree with this comment. The necessity of why the grinded samples should be separated into two parts for spectral and content measurements has been explained on page 4, lines 153 - 164 of the paper, with the following modifications:

When making measurements of soil spectral properties and iron oxide content, since these two measurements usually require the use of different instruments, the operation of which may cross-contaminate or interact with the samples, separating the samples for measurement is an effective way to avoid interference. In addition, in order to ensure the accuracy of the experimental results, the ground soil samples will be thoroughly mixed to ensure homogeneity so that even if the samples are divided into two for different purposes, the impact on the final experimental results will be small. Based on these considerations, we chose to split each soil sample after grinding into two parts: one for the measurement of the hyperspectral data of the soil and the other for the measurement of the iron oxide content in the soil. This experimental design helps to improve the accuracy of the measurements and the reliability of the experimental results.

Comments 4: On page 4, lines 174-175, it is mentioned, ‘138 soil iron oxide content data were excluded from the outliers,’ but this phrasing might cause misunderstanding. Please clarify the total number of data points and how many outliers were actually excluded. Without clear explanation, readers might misunderstand that all 138 samples were removed. It is recommended to specify both the total data count and the number of excluded outliers to avoid confusion.

Response 4: Thank you for pointing this out. We agree with this comment. The description of outlier exclusion in the paper did have a wording problem, and the description has been modified in the corresponding section. On page 5, lines 189-194, with the following modifications:

Furthermore, in order to prevent the negative impact on the modeling effect due to the existence of outliers in the iron oxide content data, an outlier elimination operation was performed, and the box-and-line plot method can effectively identify and handle outliers, and three outliers were excluded by setting plus or minus 1.5 times of the quartile spacing as the judgment criterion, as shown in Figure 2. 

Comments 5: The manuscript mentions a sample size of 138, but machine learning algorithms like RF or XGBoost typically require a large amount of labeled data. Given the relatively small dataset, this raises concerns about the reliability of the results. I recommend increasing the sample size to better meet the data requirements of machine learning algorithms. Additionally, I suggest incorporating a comparison with linear regression methods to highlight the advantages of using machine learning algorithms in this study and to enrich the results and analysis section.

Response 5: Thank you for your valuable comments. The sample collection time is early; increasing the sample for the non-same batch of data collection will affect the experimental results, so will pay attention to the sample size in future studies. In order to highlight the superiority of the machine learning algorithm model, the linear partial least squares regression model was re-added on page 12, lines 408-416 of the paper, with the following modifications:

The PLSR model inversion results are shown in Table 4. Among the PLSR models, the accuracy of the validation set of the RL-CC-CARS-PLSR model is the highest among the models constructed by using the CC-CARS algorithm to screen the characteristic wavelengths, in which the , RMSEV, and RPD of the validation set are 0.720, 7.019, and 1.891, respectively, and the accuracy of the validation set of the CR-CC-Boruta-PLSR model is the highest among the models constructed by using the CC-Boruta algorithm to screen the characteristic wavelengths. The Boruta-PLSR model has the highest accuracy, where the , RMSEV and RPD of the validation set are 0.613, 8.258, and 1.607, respectively. The two best PLSR models have RPDs between 1.4 and 2, and the models have medium predictive power.

Table 4. Results of PLSR inversion for iron oxide content in soil

CC

Wavelength selection

quantities

calibration set

validation set

RMSEC

(g/kg)

RMSEV

(g/kg)

RPD

OR

CC-CARS

27

0.687

6.002

0.375

10.490

1.265

CR

52

0.625

6.564

0.640

7.968

1.666

RL

62

0.816

4.603

0.720

7.019

1.891

SNV

68

0.680

6.066

0.404

10.243

1.296

FD

53

0.510

7.508

0.223

11.694

1.135

OR

CC-Boruta

11

0.486

7.689

0.176

12.049

1.101

CR

70

0.545

7.236

0.613

8.258

1.607

RL

13

0.550

7.192

0.347

10.722

1.238

SNV

30

0.329

8.786

0.414

10.155

1.307

FD

53

0.587

6.894

0.585

8.552

1.552

Comments 6: Following the previous question, what specific spectral bands were used in this study? Additionally, which parameters were selected? These critical details should be clearly explained in the paper to help readers better understand the rationale behind the data selection and parameter settings. I recommend elaborating on the choice of spectral bands and parameters in the methods section to ensure the transparency and reproducibility of the research process.

Response 6: Thank you for pointing this out. We agree with this comment. The specific bands selected by the feature selection method are further described in Section “3.4 Selection of characteristic bands” on page 6, lines 231-235. In addition, key details such as the parameter settings of the two characteristic bands variable preference methods are discussed, and the specific modification results are as follows:

  • (Competitive adaptive reweighted sampling (CARS) on page 6, lines 232-236) Due to the instability of the algorithm, in this study, the algorithm was run for 50 repetitions, and the frequency of occurrence of each wavelength was counted. Eventually, those wavelengths with more than 20 frequencies were selected as the characteristic wavelengths to ensure the reliability of the model, and the method was implemented through Matlab software.
  • (The Boruta algorithm (Boruta) on page 6, lines 242-247) In this paper, the algorithm is implemented in Python using the 'Brotutapy' package, and the algorithm parameters are set as follows: 'n_estimators' is set to 'auto' to automatically select the number of estimators; 'perc' was set to 95 to determine the threshold of feature importance; 'alpha' was set to 0.05 for hypothesis testing; and'max_iter' was set to 500 to specify the maximum number of iterations.
  1. Response to Comments on the Quality of English Language

Point 1:  NO

Response 1: Changes were made to address grammatical issues in the paper accordingly.

  1. Additional clarifications

The following areas have been added to this revision:

(1)   Changes were made to the title of the paper

(2)   The second part of the introduction presents in tabular form the methods and precision used in some of the most recent research advances in the field to date.

(3)   A linear partial least squares model was added to the paper to compare with a non-linear machine learning algorithm model to highlight the superiority of machine learning algorithms.

(4)   Key details such as parameterization of the two characteristic bands' variable preference methods are discussed.The parameter optimization methods and principles of the three models are discussed.

Round 2

Reviewer 2 Report

Comments and Suggestions for Authors

After reviewing the revised manuscript, I find that the authors have made improvements compared to the previous version. Their responses to the review comments are satisfactory, and the overall structure, and content quality of the paper have been enhanced. Therefore, I have no further comments and recommend the paper for direct acceptance.